# Identification and Expression Analysis of Phosphatidylinositol Transfer Proteins Genes in Rice

**DOI:** 10.3390/plants12112122

**Published:** 2023-05-26

**Authors:** Mengtian Pei, Xuze Xie, Baoyi Peng, Xinchi Chen, Yixuan Chen, Ya Li, Zonghua Wang, Guodong Lu

**Affiliations:** 1State Key Laboratory of Ecological Pest Control for Fujian and Taiwan Crops, Fujian Agriculture and Forestry University, Fuzhou 350002, China; 15870651077@163.com (M.P.); xxz2020phd@163.com (X.X.); pbypeace@126.com (B.P.); c2231668253@163.com (X.C.); cyxxuaner@163.com (Y.C.); liya-81@163.com (Y.L.); zonghuaw@163.com (Z.W.); 2Haixia Institute of Science and Technology, Fujian Agriculture and Forestry University, Fuzhou 350002, China; 3Institute of Oceanography, Minjiang University, Fuzhou 350108, China

**Keywords:** *Oryza sativa*, PITPs, innate immunity, Me JA and SA

## Abstract

The family of phosphatidylinositol transfer proteins (PITPs) is able to bind specific lipids to carry out various biological functions throughout different stages of plant life. But the function of PITPs in rice plant is unclear. In this study, 30 PITPs were identified from rice genome, which showed differences in physicochemical properties, gene structure, conservation domains, and subcellular localization. The promoter region of the *OsPITPs* genes included at least one type of hormone response element, such as methyl jasmonate (Me JA) and salicylic acid (SA). Furthermore, the expression level of *OsML-1*, *OsSEC14-3*, *OsSEC14-4*, *OsSEC14-15*, and *OsSEC14-19* genes were significantly affected by infection of rice blast fungus *Magnaporthe oryzae*. Based on these findings, it is possible that OsPITPs may be involved in rice innate immunity in response to *M. oryzae* infection through the Me JA and SA pathway.

## 1. Introduction

Since lipids are the main components of eukaryotic cell membranes and are also important constituents of bacteria cell walls, they play a significant regulatory role in membrane transport and signal transduction. Phosphatidylinositol transfer proteins (PITPs) are indispensable for the growth and development of eukaryotes, because PITPs can exert specific biological functions by binding the related lipids in different cellular processes. According to their different structures, PITPs can be divided into 4 classes, namely the Class I PITPN, the Class II RDGB, the Class III SEC14 and the Class IV ML family.

### 1.1. PITPs in Animals

The Class I PITPN and Class II RDGB are widely distributed in animals. The PITP domain they contained can bind with different lipids including phosphatidylinositol (PI) and phosphatidylcholine (PC), and promote the mutual transport of lipids between the cellular endomembrane systems [1]. Class I PITPN primarily includes PITPα (PITPNA) and PITPβ (PITPNB). They are soluble, single-domain proteins with a relative molecular weight of approximately 35 kD, and contain one PITP domain which basic biological activity is lipid exchange [2]. The loss of *PITPα* often leads to neurodegenerative diseases and liver dysfunction in animals [3,4]. PITPβ has 77% sequence identity with PITPα, and it is mainly involved in the retrograde transport of lipids from the Golgi apparatus to the endoplasmic reticulum [5]. In animals, Class II RDGB includes the multi-domain RdgBα and the single-domain RdgBβ. They exhibit significant PI transport activity and play an irreplaceable role in maintaining lipid stability and signal transduction in cells [6]. As the earliest discovered RdgB protein, DmRdgBα is an important component for maintaining visual transduction of drosophila by maintaining the retina ultrastructure [7]. Apart from the PITP domain, RdgBα also contains the FFAT (two phenylalanines in an acidic tract) domain, the DDHD (dua1-dialyzerhemodi-alysis) metal-binding domain, and the LSN2 (large scale network address translation 2) domain [8]. While RdgBβ only contains one PITP domain, and it is only found expressed in multicellular animals. In addition to transporting PI, RdgBβ can also bind and transport phosphatidic acid (PA) in large quantities [9]. Furthermore, both RdgBα and RdgBβ can maintain the stability of intracellular endomembrane system and normal function of second messengers by transporting PA [10].

The Class III SEC14 does not contain the PITP domain, and as a replacement, its SEC14 domain can complete the duty of lipid transfer [11]. The yeast Sec14p is the first member of Sec14 protein family, subsequently, homologs of this protein have been extensively cloned and studied in animals [12]. In addition to possessing a canonical lipid activity transfer, the Sec14 protein is associated with tumors, cancer, and innate immunity in animals. For example, the expression level of human *SEC14L1* is positively correlated with the degree of invasive breast cancer, and is therefore considered an independent prognostic indicator for breast cancer [13,14]. The antiviral signal transduction pathway mediated by Retinoic acid-inducible gene I (RIG-I) can be negatively regulated by the protein encoded by *SEC14L1*, through inhibition of interaction between RIG-I and downstream effectors [15]. Moreover, both human *SEC14L2* and *SEC14L3* exhibit an active role in suppressing tumor growth by exerting anti-proliferative effects [16,17].

The Class IV ML has been extensively researched in both mammals and insects. The β-rich-sheets predicted within their ML domains are believed to consist of multiple chains and facilitate various biological activities through their interactions with specific lipids [18]. MD-2 is the initial ML discovered in a human kidney cell line, serving as an extracellular binding chaperone for Toll-like receptor 4 (TLR4) [19]. The combination of MD-2 and TLR4 leads to enhance recognition and binding of lipopolysaccharide (LPS), a component of bacterial cell wall, thereby inducing the expression of immune response-activating cytokines interleukin 10 (IL-10) and transforming growth factor-β 1 (TGF-β 1) [20,21,22]. Numerous investigations have indicated the involvement of ML protein in the innate immunity process. The activity of binding LPS, has been observed in MD-2 of *Mus musculus*, MsMl-1 of *Manduca sexta*, and LvMl of *Litopenaeus vannamei*, suggesting that ML recognizes LPS signal and takes part in the innate immunity process [23,24,25]. The AgMdl-1 homolog of *Anopheles gambiae* plays a role in the immune defense of mosquitoes against *Plasmodium falciparum* infection [26]. The ML domain-containing protein BmEsr16 from the silkworm (*Bombyx mori*) has the ability to bind to a range of bacterial cell wall components, including LPS, lipid A, peptidoglycan (PGN), and lipoteichoic acid (LTA). And loss of function of BmEsr16 can result in a reduction of antibacterial response [27,28]. Furthermore, the salivary glands and hemolymph of castor seed ticks (*Ixodes ricinus*) express the *IrML* gene throughout all developmental stages, indicating its potential involvement in innate immune responses [29].

### 1.2. PITPs in Plants

So far, none of Class I and II PITPs has been reported in plant, whereas, the class III SEC14 is well known with wide range of homologous proteins.

The Sec14 protein family performs a variety of roles in plants, encompassing osmoregulation, cell polarity growth, nodule formation, protein conveyance, regulation of plant immune response, and interaction with viruses [30,31]. Based on their various domains, the Sec14 proteins can be categorized into three types, Sec14-Only protein, Sec14-Nodulin protein, and Sec14-GOLD protein. Sec14-Only protein possesses only one SEC14 domain and plays a crucial role in plant stress response. For instance, *HvSEC14p* in barley (*Hordeum vulgare*), *TaSEC14-5* in wheat (*Triticum aestivum*), *ScSEC14-1* in sugarcane (*Saccharum officinarum*), and *ZmSEC14* in maize (*Zea mays*) were significantly upregulated under osmotic stress, indicating that SEC14 can enhance transport activity of PI and help maintain stability of cell membranes [32,33,34]. After being infected by *Ralstonia solanacearum*, the expression of *NbSEC14* gene in tobacco (*Nicotiana tabacum*) was significantly increased. Meanwhile, when this gene was silenced, the defense-related gene *PR-4* was down-regulated, and there was an accumulation of jasmonate (JA) and methyl jasmonate (Me JA), suggesting that Sec14 is strongly associated with the immune response of plants [35,36]. The Sec14-Nodulin and Sec14-GOLD protein possess a SEC14 domain, as well as a Nodulin domain and a GOLD domain at the C-terminal, respectively. Sec14-Nodulin protein primarily serves in membrane morphogenesis and root hair development, while Sec14-GOLD protein is mainly involved in cell division [37,38]. As an example, AtSfh1 protein in *Arabidopsis thaliana* is a Sec14-Nodulin protein that is concentrated around the non-continuous area of the plasma membrane and cytoplasm at the extremity of root hairs. When AtSfh1 function is lost, it will cause a breakdown in the polarity of root hairs in *A. thaliana*, leading to incorrect extension [39,40]. In the process of cell division, AtPatl2 (PATELLIN2), the Sec14-GOLD protein in *Arabidopsis*, works in cooperation with mitogen-activated protein kinase (MAPK) to facilitate cell plate formation by adjusting the binding strength of phosphoinositide at the division plane [41].

In plant, The Class IV ML is expected to participate in the transport of sterols, as indicated by the Gene Ontology term GO:0032366 (http://amigo.geneontology.org/amigo/term/GO:0032366, accessed on 23 February 2023). Additionally, the CDD website of NCBI suggests that the ML domain of this protein may also function as a PG-PI_TP domain, facilitating the transportation of phosphatidylinositol and phosphatidylglycerol (https://www.ncbi.nlm.nih.gov/Structure/cdd/cddsrv.cgi?uid=238459, accessed on 26 February 2023). In *A. thaliana*, AtRosy1 (AT2G16005) contains the ML domain and is a gravity-responsive protein found in the root tip. It interacts with stigmasterol and ethanolamine phosphate (stigmasterol and phosphoethanolamines) and influences the gravitropism of roots [42]. ML are involved in innate immunity in plants. In *A. thaliana*, the expression of *AtML3* (AT5G23820) is increased in response to physical damage and induction by jasmonic acid JA, indicating that it is associated with the plant immune response. The loss of *AtML3* in *A. thaliana* results in reduced disease resistance, further supporting the role of *AtML3* in plant immunity [43]. The cuticle plays a vital role in protecting plants against dryness, diseases, and pests, serving as a surface barrier between them and their environment. Recent studies have identified an *ML* gene in the epidermis of tomato (*Solanum lycopersicum*), with higher expression levels in this tissue than in others. Early growth stages of tomato also exhibited high transcriptional activity of this protein. It is hypothesized that the ML domain of this protein is capable of binding to pentacyclic triterpenes abundant in tomato epidermis wax, and thus, contributing to the biosynthesis, assembly, and recombination of the cuticle [44].

Rice is among the most significant staple foods, feeding nearly half of the world’s population. Despite significant efforts directed towards cultivating highly productive and multi-resistant rice varieties, the impact of diseases continues to cause substantial annual reductions in rice yield [45]. Due to its fixed growth environment and singular planting pattern, rice plant is susceptible to various diseases and insects. To cope with this stress, rice has evolved complex and systematic resistance mechanisms. The phosphatidylinositol transfer proteins carry out various biological functions including innate immunity in animal and plant. But the function of PITPs in rice plant is unclear. In this paper, a total of 30 rice PITPs were identified and their motif composition, protein structure, phylogeny, and cis-acting elements in gene promoter were characterized. Additionally, the expression of PITPs gene was detected during *Magnaporthe oryzae* infection, to investigate the involvement of these genes in rice resistance. We also observed the protein localization of the highly expressed genes. Overall, our study provides a theoretical foundation for further exploration of the function of PITP family in rice immunity.

## 2. Results

### 2.1. Identification and Bioinformatics Analysis of PITP Genes in Rice

As the Class I PITPN and Class II RDGB have only been reported in humans and animals, we conducted a homologous search in the rice genome for *PITPα* and *PITPβ* of mouse Class I PITPN, and *RDGBα* of human Class II RDGB. However, result shown that there was only a single homologous gene, namely *OsPITPα*.

Furthermore, utilizing Class III *ScSEC14* in *S. cerevisiae* and Class IV *AtROSY1* in *A. thaliana*, we identified 27 *OsSEC14* and 2 *OsML* homologous genes in rice. These 30 *OsPITPs* were named according to their location on the chromosome. And we were unable to locate related information on the *OsSEC14-25*, *OsSEC14-26*, and *OsSEC14-27* genes in NCBI. However, we were able to find these genes in the rice UGA database (http://rice.uga.edu/cgi-bin/gbrowse/rice, accessed on 7 February 2023). It is possible that these three genes are either low expression or pseudogenes, given the lack of information in NCBI. As a result, we assigned separate names to each of these three genes (Table 1). Through bioinformatics analysis, it was found that OsPiptα, OsSec14, and OsMl proteins exhibit significant differences in their physicochemical properties. The number of amino acids in these proteins ranged from 152 to 671, with molecular weights ranging from 16196.64 to 75911.99 Da. The theoretical isoelectric point ranged from 4.59 (OsSec14-15) to 9.95 (OsSec14-26), with an average of 8.2375. Among the identified proteins, 16 were basic proteins (pI > 7.5) and 14 were acidic proteins (pI < 7.5). Furthermore, 73% of the proteins were unstable (25 out of 30, defined as having an instability coefficient greater than 40), with fat coefficients ranging from 68.71 to 96.86. Hydrophilic proteins accounted for 90% (27 out of 30, defined as having a GRAVY score less than 0), while hydrophobic proteins accounted for 10% (3 out of 30, defined as having a GRAVY score greater than 0), indicating that the family members were mainly hydrophilic proteins. The predicted subcellular localization of OsPitps suggested that they were mainly located in the chloroplast, cytoplasm, mitochondria, and nucleus, except OsMl-2 with extracellular localization (Table 2).

### 2.2. Evolution Analysis of PITPs in Plants

To investigate the evolutionary relationship of rice PITPs family, the phylogeny of OsPitpα, OsSec14, and OsMl proteins were analyzed as shown in Figure 1. The family members were classified into five groups, namely groups I, II, III, IV, and V. The OsPitpα, OsMl1, and OsMl2 formed a separate branch (I). 27 members of OsSec14 subfamily were assigned to groups II, III, IV, and V based on the phylogenetic relationship, and group V had the most members (11), followed by group II (7), while group IV had the fewest members (3), indicating that there were differences in the number and homology of PITPs family proteins during the evolutionary process.

To investigate the phylogenetic relationship between rice PITPs family proteins and other crops such as maize, barley, and *A. thaliana*, we obtained 1, 1, and 0 OsPitpα homologous proteins, 92, 35, and 32 OsSec14 homologous proteins, and 5, 9, and 9 OsMl homologous proteins from maize, barley, and *A. thaliana*, correspondingly. By utilizing MEGA7.0 software, we conducted a cluster analysis, and the result was displayed in Figure 2. According to the genetic relationship of the 214 proteins in the phylogenetic tree, they were divided into six groups: I, II, III, IV, V, and VI, and there were variations in the species and quantity of species in different groups. OsPitpα protein and OsMl protein were in the same branch (VI) in the evolutionary tree. Rice OsSec14 protein did not belong to group VI and was different from OsSec14 protein in maize, barley, and *A. thaliana*. OsPitpα protein was primarily distributed in groups I and II, containing 8 and 7 OsSec14 proteins, respectively. These findings suggest that there are differences in homology between and within species during the evolution of PITPs. These distinctions may be related to the diverse functions of PITPs in plants.

### 2.3. Analysis of Collinearity between PITPs

Organisms replicate genes to create new family members, thereby improving gene structure and increasing functional diversity [46]. Gene replication can be broadly classified into three categories: genome-wide replication events, fragment replication events, and tandem repeat [47]. To investigate the expansion pattern of rice *PITP* genes, intraspecific collinearity analysis was conducted on *OsPITPα*, *OsSEC14* and *OsML* genes. The analysis revealed that 9 *OsSEC14* and 2 *OsML* genes formed 6 pairs that exhibited collinear relationship, while the others did not. Among these gene pairs, *OsSEC14-23* showed the highest collinearity with both *OsSEC14-21* and *OsSEC14-9* genes, and was in fact the most collinear gene among all members of this gene family. Other gene pairs that exhibited collinear relationship were OsML1 and OsML2, *OsSEC14-7* and *OsSEC14-17*, *OsSEC14-2* and *OsSEC14-16*, and *OsSEC14-3* and *OsSEC14-1* (Figure 3). Additionally, we found that among the 6 pairs of collinear genes, 2 *OsML* and 9 *OsSEC14* genes underwent segment replication, suggesting that the gene family improved its gene structure and functional diversity through fragment replication.

To gain a deeper understanding of the expansion pattern of the PITPs family in *Arabidopsis*, wheat, and tomato, we conducted collinearity analyses of *OsPITPα*, *OsSEC14*, and *OsML* with the respective genomes of these plants. As shown in the Figure 4, the findings revealed that there was no discernible collinear relationship between *OsML* and the *Arabidopsis* genome. However, *OsSEC14* and *Arabidopsis*
*AtSEC14* exhibited collinear relationship, with five pairs of genes forming collinear relationships. The collinear analysis of *OsPITPα*, *OsSEC14* and *OsML* genes with wheat genomes resulted in the formation of 75 pairs of homologous genes between the two species. The rice *OsPITPα*, *OsSEC14* and *OsML* genes exhibited homologous correspondence with multiple genes in wheat genome, with *OsML1*, *OsSEC14-2*, *OsSEC14-3* and *OsSEC14-15* showing the most corresponding relationships with wheat genes (6). On the other hand, nine genes including *OsPITPα*, *OsSEC14-4*, *OsSEC14-5*, *OsSEC14-9* and *OsSEC14-26* had the least corresponding relationship (3), while other genes had 5 corresponding relationships with wheat genome, such as *OsML2* and *OsSEC14-17* genes. These findings suggest that members of the wheat *PITPα*, *SEC14* and *ML* genes family are homologous repeats of the *OsPITPα*, *OsSEC14* and *OsML* genes in rice. The collinear analysis of tomato revealed five homologous gene pairs, all of which were wheat *OsSEC14* genes, consistent with the collinear analysis of *A. thaliana*. Notably, we found that there was no collinear relationship between wheat *OsML* genes and *A. thaliana* or tomato genomes, which is consistent with species evolution.

### 2.4. Examining the Conserved Motifs and Gene Structure of OsPITPs

The conserved motifs of OsPitps were predicted, and their base sequences were analyzed, which showing differences in the number and type of motifs contained (Figure 5a). The length of motif is also varied, with OsPitpα containing only motif 10. OsMl contains motif 6 and motif 7, both of 4aa8 and 49aa lengths, while the 27 OsSec14 proteins (except OsSec14-27) contain motif 2 and motif 3, with lengths of 29aa and 31aa, respectively (Figure 5b). This indicates that although they are members of the OsPITPs family, they contain different types of motifs, suggesting specific biological functions. Furthermore, 11 proteins including OsSec14-2, OsSec14-16, and OsSec14-5 contain five motifs (motif 1–5), OsSec14-4, OsSec14-7, OsSec14-12, and OsSec14-22 contain three motifs (motif 2, 5, and 3), 10 proteins such as OsSec14-3, OsSec14-14, and OsSec14-15 contain two motifs (motif 2, 3), while OsSec14-27 contains only motif 2. These findings demonstrate that proteins of the same type combine different motifs and play different functions in organisms.

Through the examination of the gene structure of *OsPITPs* (Figure 5c), it was discovered that there were differences in the number of exons present in *OsPITPα*, *OsSEC14* and *OsML* genes. *OsSEC14-27* contained the fewest exons, totaling 3, while *OsML1*, *OsML2*, *OsSEC14-3*, and *OsSEC14-15* contained 4 exons. Meanwhile, *OsSEC14-2* had the most exons, with 15 present, and other genes had exons ranging from 5 to 12. The uneven distribution of exons suggests that intron insertion or deletion may have taken place during the evolution of *OsPITPs*.

### 2.5. The Functional Domains of PITPs Family Proteins

Based on the predicted functional domains of PITPs family proteins, it was discovered that OsPitpα contains an IP_trans domain, while both OsMl have a ML domain, and 27 OsSec14 have a SEC14 domain (Figure 6). These domains have the ability to bind to specific lipids and carry out specialized biological functions. Additionally, all 24 OsSec14 contain an unknown functional domain, which is the N-terminal domain of CRAL-TRIO. This domain is a common characteristic among various lipid-binding proteins and is capable of binding to small lipophilic molecules (http://www.ebi.ac.uk/interpro/entry/InterPro/IPR001251/, accessed on 13 February 2023). Furthermore, OsSec14-2, OsSec14-16, and OsSec14-20 have transmembrane regions at their C-terminal. OsMl contains a signal peptide. It is speculated that these regions enable them to be located on the cell membrane or secrete to the extracellular space and participate in the recognition and transport of certain lipids.

### 2.6. Predicting Cis-Acting Elements of PITPs Family Genes

The upstream 2.0 kb promoters of *OsPITPα*, *OsSEC14* and *OsML* genes were analyzed for their cis-acting elements. Figure 7 shows that the promoters of *OsPITPα*, *OsSEC14* and *OsML* contain three types of regulatory elements: growth and development, hormone regulation, and abiotic stress. The growth and development type includes regulation of endosperm, protein metabolism, meristem, seeds, and circadian rhythm. Hormone regulation comprises cis-acting elements for jasmonic acid, abscisic acid, salicylic acid, auxin, and gibberellin regulation. Abiotic stress elements include light, hypoxia, anaerobic, low temperature, drought, trauma, defense, and stress regulatory elements. Moreover, there were varying types and numbers of regulatory elements found in the promoters of different *OsPITPα*, *OsSEC14* and *OsML* genes, indicating that they may be involved in plant response to abiotic stress and hormone regulation through multiple cis-acting elements, and regulate rice growth and development through different cis-acting elements.

### 2.7. Expression Levels of PITPs Genes during Infection of Rice Blast Fungus

Rice is the world’s most significant food crop, responsible for sustaining over half of the global population. However, rice blast, caused by the filamentous ascomycetes *M. oryzae*, is the most devastating fungal disease that affects rice crops [48]. Rice blast is responsible for a yearly grain loss of approximately 3 billion kilogram in China alone, whereas the loss of grain production worldwide is enough to feed 60 million people [49]. The transcriptome data of Nipponbare (NPB, a susceptible rice variety) after 24 h and 48 h *M. oryzae* infection was collected to explore whether PITPs family genes could potentially be involved in disease resistance. To analyze the gene expression level, we calculated the multiple of the gene expression level at 24 h and 48 h relative to the gene expression level at 0 h. A multiple greater than or equal to 1.2 was classified as up-regulation, whereas a multiple less than or equal to 0.8 was labeled as down-regulation. The multiple between 0.8 and 1.2 was categorized as the same level of gene expression.

According to the results, as Figure 8 illustrates, the *M. oryzae* infection led to up-regulation of the expression of 14 genes, concerning *OsML-1* and *OsSEC14-4* among others. Concurrently, eleven genes, such as *OsSEC14-27* and *OsSEC14-16* had down-regulated expression levels during the same process, whilst the gene expression levels associated with three genes such as *OsSEC-2* remained unchanged. Furthermore, the expression levels of *OsSEC14-26* and *OsSEC14-14* had down-regulated originally but then became up-regulated during the *M. oryzae* infection. Consequently, 8 genes with the highest expression levels were selected for qRT-PCR analysis to confirm the changes in gene expression during the *M. oryzae* infection. The results showed that the expression of *OsML-1* increased significantly from 24 h to 48 h after infection, while the expression of *OsML-2* decreased during the same time (Figure 9). These results suggest that *OsML-1* and *OsML-2* might have different functions in plant defense. Additionally, the expression of *OsSEC14-3*, *OsSEC14-4* and *OsSEC14-15* were increased significantly during fungal infection. In contrast, *OsSEC14-19* was rapidly down-regulated at 12 h and slightly up-regulated between 24 h and 48 h with a considerable difference in expression levels (Figure 9). Hence, we speculate that *OsML-1*, *OsSEC14-3*, *OsSEC14-4*, *OsSEC14-15*, and *OsSEC14-19* may play important roles in rice immunity.

### 2.8. Subcellular Localization of PITP Family Proteins

Based on the differences in gene expression levels, we picked out six proteins (OsMl-1, OsMl-2, OsSec14-3, OsSec14-4, OsSec14-15, and OsSec14-16) for transient expression in rice protoplasts. Our findings (as illustrated in Figure 10) revealed that OsMl-1 may be located in the cytoplasm, OsMl-2 located in the cytoplasm and cell membrane. whereas OsSec14-3 was found in the cell membrane and nucleus, which differed from our prediction in Table 2. OsSec14-4 may be located in Golgi and cytoplasm, while OsSec14-15 were located in the cytoplasm, OsSed14-16 were located in the cell membrane. It is speculated that they perform different functions in the organism.

## 3. Materials and Methods

### 3.1. Identification of Rice PITP Gene Family Members

The Rice Genome Annotation Project (RGAP) website (http://rice.uga.edu/cgi-bin/gbrowse/rice, accesed on 7 February 2023) were used to conduct a search for the *PITPα* and *PITPβ* gene from mice, *RDGBα* gene from humans, *ScSEC14* gene from yeast, and *AtROSY1* gene from *Arabidopsis*, belonging to the Class I, II, III and IV PITPs respectively. The outcome of this search identified 1 *OsPITPα*, 2 *OsML*, and 27 *OsSEC14* genes [50].

### 3.2. Physicochemical Properties and Subcellular Localization Analysis

The physicochemical properties of proteins encoded by PITP family genes, including amino acid count, molecular weight, theoretical isoelectric point, instability index, aliphatic index, and hydrophilicity, were analyzed using the Protparam program available on the EXPASy website (https://web.expasy.org/protparam, accesed on 8 February 2023) [51]. The Plant-mPLoc website (http://www.csbio.sjtu.edu.cn/bioinf/plant-multi/, accesed on 7 April 2023) and the CELLO website (http://cello.life.nctu.edu.tw/, accesed on 8 February 2023) were utilized for conducting the subcellular localization prediction [52,53]. 

### 3.3. System Evolutionary Analysis

The neighbor-joining method was utilized in constructing a systematic phylogenetic tree through the MEGA7.0 software [54]. Subsequently, the phylogenetic tree underwent modification and aesthetic enhancement using the Evolview website (http://www.evolgenius.info/evolview/, accesed on 6 March 2023) [55].

### 3.4. Collinearity Analysis

The BLASTP e-values and MCScanX software were utilized to assess the fragment and tandem duplications [56]. The TBtools program v1.098 was employed to visualize the chromosome location and duplicated genes [57].

### 3.5. Gene Structure and Protein Motif Analysis

Identify conserved motifs through the MEME website (http://meme-suite.org/tools/meme, accesed on 10 February 2023) [58]. Determine regulatory elements through the PlantCARE website (http://bioinformatics.psb.ugent.be/webtools/plantcare/html, accesed on 10 February 2023) [59], and visualize the composition of gene structure and cis-acting elements using TBtools v1.098 software.

### 3.6. Functional Domain Analysis

The SMART website (https://smart.embl.de/, accesed on 10 February 2023) was used to forecast the functional domain of OsPITP proteins [60].

### 3.7. Subcellular Localization

The coding sequences (CDS) for the *OsML* and *OsSEC14* genes in rice were acquired from the rice UGA database. RFP fusion proteins were then created using the pHF225 vector and introduced to rice protoplasts. Lastly, confocal microscopy at a magnification of 60× was used to observe the subcellular localization of the produced proteins.

### 3.8. Transcriptome Data

Nipponbare (NBP) plants were inoculated with Guy11 strain of *M. oryzae* and samples were collected at 0, 24, and 48 h post-infection. Control samples were also collected from uninfected NBP plants at the same time points. Subsequently, all samples were flash-frozen in liquid nitrogen and sent to BioMarker Technologies (Beijing, CHN)for transcriptome sequencing analysis.

### 3.9. RNA Extraction and qRT-PCR

ZH11 rice seedlings at 3–4 weeks old were chosen for the experiment. They were subjected to spray-inoculation with the rice blast fungus at four different time points: 0 h, 12 h, 24 h, and 48 h. Total RNA from rice leaves was then extracted using the RNA extraction kit (Eastep^®^ Super Total RNA Extraction Kit, LS1040, Promega, Fitchburg, MA, USA) and reverse transcription was conducted utilizing the reverse transcription kit (1st Strand cDNA Synthesis SuperMix for qPCR, 11141ES60, Yeasen Biotechnology, Shanghai, China) to convert the extracted RNA into cDNA. Eventually, qRT-PCR was conducted on rice *OsML* and *OsSEC14* genes using the qRT kit (qPCR SYBR Green Master Mix, 11201ES08, Yeasen Biotechnology, Shanghai, China).

The extraction of total RNA was completed by the RNA extraction Kit (Easter^®^ Super Total RNA Extraction Kit, LS1040, Promega, Fitchburg, MA, USA). This kit has a unique cell lysis system, using a centrifuge column containing a silicon matrix membrane to adsorb nucleic acid molecules, and then removes genomic DNA contamination through DNA enzymes to obtain high-purity total RNA. Evaluate the quality of RNA by observing its A260/A230 (2.1–2.6) and A260/A280 (1.9–2.1).

The reverse transcription of RNA was completed by the RNA reverse transcription kit (Evo M-MLVRT Mix Kit With gDNA Clean for qRCR Ver.2, AG11728, Accurate Biology, Huanan, CHN). This kit contains gDNA Clean Reaction Mix Ver.2, which can efficiently remove genomic DNA from RNA and optimize all components required for reverse transcription reaction. Its reverse transcript product can be directly subjected to RT-PCR.

Gene expression analysis was performed using the qRT kit (SYBR Green Premix Pro Taq HS qPCR Kit, AG11701, Accurate Biology, Huanan, CHN), qRT instruments (Pangaea, Pangaea 6, Apex Biotechnology, Suzhou, China), and qRT software (CqMANs). The primer design and reaction cycle are shown in Table 3, and the RT-PCR data uses 2^−ΔT^ to represent gene expression levels [61].

## 4. Discussion

Plants are susceptible to biological stress caused by viruses, bacteria, fungi, insects, and other pathogens, as well as abiotic stress such as high temperature, freezing, drought, waterlogging, and starvation due to the uncontrollable natural environment and fixed growth locations. PITPs are crucial for lipid transport and signal transduction, which can maintain cell membrane stability and enable plants to respond quickly to stress. PITPs have distinct domains, where Class I and II PITPs contain PITP domains, Class III PITPs have SEC14 domains, and Class IV PITPs contain ML domains. These domains perform unique biological functions by binding to various lipids that are vital for plant development, growth, and stress response. In this study, PITPs in rice were characterized. Firstly, we found that there are 30 PITPs in rice by using homology comparisons, including 1 *OsPITPα*, 27 *OsSEC14*, and 2 *OsML*. Secondly, we used bioinformatics methods to reveal the diversity of OsPITPs in terms of physicochemical properties, evolutionary level, conserved motifs, functional structure, and cis-acting element distribution. Furthermore, we discovered that *OsPITPs* exhibit different expression patterns in the process of pathogen invasion through gene expression analysis. Finally, we found that the spatial distribution of OsPITPs also varies by observing subcellular localization. These findings provide important clues for further exploration of the biological functions of OsPITPs.

There are obvious differences in the physicochemical properties of OsPITPs in rice, and it is predicted that they are mainly located in chloroplasts, cytoplasm, mitochondria, and nuclei. These results provide some reference value for the biological function of OsPITPs. In the analysis of physical and chemical properties, most of the OsPITPs in rice are hydrophilic proteins, and only two OsMl proteins are shown to be hydrophobic proteins. This indicates that OsMl proteins may recognize and wrap lipids, and then transport them to the next location. The mechanism of action of OsMl may be similar to that of MD-2 in animals, which can wrap LPS and transport it to the vicinity of TLR4, and then be recognized and activate the immune response by TLR4 [20].

PITP family genes play an important role in plant evolution, but the number and homology of family members have differed in this process. We conducted a systematic phylogenetic analysis of OsPITPs family proteins in rice and found that OsPitpα, OsMl1, and OsMl2 exist in Branch I, while the remaining 27 OsSec14 are divided into four branches. By integrating the PITP family proteins of rice, maize, barley, and *Arabidopsis* for phylogenetic analysis, the results showed that the PITP family members of these four species were divided into six groups, each containing different numbers and species of PITPs. Further clustering analysis showed that the homology of PITP family members varied between different species and groups, with higher homology between Pitpα and Ml, and lower homology with Sec14, which may be related to different gene functions. In summary, plants may have chosen different branches of evolution to better recognize and bind specific lipids, so that each different lipid can be recognized by the plant and responded to in a timely manner.

Understanding the expansion patterns of gene families can help to deeply understand the evolutionary process and functional differences of the gene family. Therefore, we conducted intraspecific co-linearity analysis of the *OsPITPs* gene family, and the results showed that the *OsPITPs* gene family enhanced functional diversity and improved gene structure by fragment duplication, thereby obtaining greater adaptability and flexibility. In addition, we also conducted interspecific co-linearity analysis with the *PITPs* gene families of Arabidopsis, wheat, and tomato. The results showed that homologous gene pairs were found in the *PITPs* gene family of rice and wheat in the *PITPα*, *SEC14*, and *ML* genes, while only homologous gene pairs were found in the *SEC14* gene of rice and the other two species. These results indicate that the relationship between rice and wheat is closer, And *OsPITPα* and *OsML* are more conservative in evolution.

The conservation motif and functional structure analysis of OsPITPs family proteins have revealed that different proteins of the same family members may have different types and lengths of motifs, and conservation domains that bind to different lipids to perform different functions. Meanwhile, OsSec14-2, OsSec14-16, and OsSec14-20 have transmembrane domains at the C-terminus, indicating that they may be located on the cell membrane and participate in the perception and transport of certain lipids. These analytical results indicate that OsPITPs family proteins are very complex, and different proteins of the same family can have different biological functions. This functional diversity may be related to functional structural domains and protein localization.

We conducted cis-acting regulatory elements analysis of the promoter sequences of rice *OsPITPs* genes and found different types and numbers of regulatory elements, including growth and development, hormone regulation, and stress response elements. These elements may be important factors controlling the specific expression of these genes under different conditions, and together regulate the life cycle process of rice. This result provides important clues for understanding the function and regulatory mechanisms of *OsPITPs* genes, and provides a foundation for further research. However, this study is only based on bioinformatics predictions and analysis, and further experimental verification is needed to confirm the true function of these predicted regulatory elements. It is worth noting that the promoter parts of different *OsPITPs* in rice all contain at least one salicylic acid (SA) or methyl- jasmonate (Me JA) response element. Jasmonic acid (JA) and its derivatives such as Me JA are important signal molecules in plant defense reactions and are widely present in the plant kingdom. Studies have shown that jasmonic acid compounds can induce the expression of plant endogenous disease resistance genes and many substances involved in defense response material synthesis to participate in the defense response, thereby regulating plant disease resistance [62,63,64,65]. SA, as a signal molecule, plays an important role in plant immunity. Exogenous SA or increased endogenous SA levels can induce the expression of PR genes and enhance plant disease resistance [66,67,68,69]. The above results indicate that members of the rice OsPITPs family may participate in plant disease resistance through SA and JA-mediated pathways.

In order to investigate the role of OsPITPs in rice disease resistance, we examined the expression of all *OsPITPs* genes in transcriptome data of rice infected with blast fungus. We found that 14 genes were upregulated, 11 genes were downregulated, and 2 genes were initially downregulated and then upregulated. This result suggests that the expression levels of these genes changed significantly during infection and may be related to rice disease resistance. Subsequently, we selected some genes for further verification by qRT-PCR and confirmed that *OsML-1*, *OsSEC14-3*, *OsSEC14-4*, *OsSEC14-15* and *OsSEC14-19* played certain roles in the process of blast fungus infection. However, compared with the transcriptome data, only the expression trends of *OsML-1* and *OsSEC14-4* in the qRT-PCR results were consistent with the transcriptome data, emphasizing their positive role in rice disease resistance. The difference between the qRT-PCR results and the transcriptome data may be due to different sample origins. Although both are susceptible varieties, ZH11 is more resistant to disease than NBP. These results provide important information for further exploring the role of the OsPITPs family members in rice disease resistance.

To better understand the biological functions of OsPITPs, we examined the localization of six highly expressed OsPITPs proteins and found that their localization differed from the predicted localization results in Table 1. Specifically, OsMl-1 and OsMl-2 were localized to the cytoplasm, OsSec14-3 was localized to both the cytoplasm and nucleus, OsSec14-4 was localized to the Golgi apparatus and cytoplasm, while OsSec14-15 and OsSec14-16 were localized to the cytoplasm. These results indicate that there are differences between the predicted localization results in the table and the actual localization, and further experiments are necessary to confirm the protein localization. Accurate protein localization is essential for further understanding its function and involvement in biological processes.

In this study, one of our main focuses is to investigate the role of OsPITPs in rice immune response. The promoter region of rice *OsML-1* contains SA hormone response elements, and the gene expression of *OsML-1* is continuously upregulated during the blast fungus infection, suggesting that *OsML-1* may be regulated by SA hormone in rice immune process. Although plants cannot perform behaviors like animals do to maximize benefits and avoid harms, they have evolved a complex system of recognition and signal transduction to respond to environmental stressors [70,71,72,73]. Receptor-like kinases (RLKs) and receptor-like cytoplasmic kinases (RLCKs) play important roles in this signal recognition and transduction system, and are involved in sensing external stimuli, activating downstream signal pathways, and regulating immune responses [74]. For example, in plant immunity, *Arabidopsis* RLK protein FLS2 and EFR bind to bacterial flagellin protein (flg22) and elongation factor (elf18), respectively, recruit and activate RLCK protein BAK1 and SERK3, then form an active receptor complex to transmit the signal to the cytoplasm [75,76]. Interestingly, we found that rice OsMl-1 can interact with RLCK OsBsr1 (unpublished data). As a molecule of RLK, RLCK OsBSR1 can mediate chitin-involved rice cell defense signal pathway and is regarded as a broad-spectrum disease resistance gene [77,78]. We speculate that the protein encoded by OsML-1 may recognize the lipid component of the pathogenic fungus cell wall, transmit the signal to OsBsr1, and trigger rice immune response.

Among the 27 *OsSEC14* genes in rice, the expression levels of *OsSEC14-3*, *OsSEC14-4*, *OsSEC14-15*, and *OsSEC14-19* genes were significantly up-regulated or down-regulated during the initial infection stage of blast fungus, indicating that these genes are involved in rice’s biotic stress response process. According to the homologous alignment and domain analysis results, OsSec14-4 and OsSec14-19 belong to Sec14-Only, while OsSec14-3 and OsSec14-15 belong to Sec14-GOLD. In *Nicotiana benthamiana*, the expression level of Sec14-Only gene *NbSEC14* was significantly upregulated after pathogen infection [35]. It has also been found that silencing of *NbSEC14* gene can lead to defective expression of defense-related PR genes and significant accumulation of plant hormone JA and its derivative Me JA [36]. Recent studies have shown that the *NbSEC14* gene may also contribute to the induction and triggering of PTI immune responses [79]. According to homologous alignment and cis-element analysis, OsSec14-4 and OsSec14-19 belong to Sec14-Only and have hormone response elements such as Me JA and SA, which implies that their role in plant immunity may be similar to the *NbSEC14* gene. Sec14-GOLD is also known as PATL protein (Patellin), which originates from the Latin word “patella”, representing small plates [38]. Interestingly, PATL proteins not only participate in cell division but also in plant immune defense responses [38]. In *Arabidopsis*, members of the PATL family, PATL3 and PATL6, can interact with the movement protein (MP) of alfalfa mosaic virus (AMV), thereby interfering with virus movement [80]. Overexpression of PATL3 and PATL6 leads to weakened virulence, and at the same time, RNA accumulation of the virus increases in PATL3 and PATL6 knockout mutants [80]. During the infection process of blast fungus in rice, a large number of effectors are released to promote the colonization of the pathogen [81,82]. *OsSEC14-3* and *OsSEC14-15*, belonging to Sec14-GOLD in rice, were significantly upregulated after 48 h of infection with blast fungus, indicating that these genes may be induced by some effectors of *M. oryzae*, thus playing a positive regulatory role in plant resistance.

This study explored the diversity of the rice OsPITPs family and their expression patterns during pathogen infection. The results showed that the OsPITPs family proteins are very complex, and different members of the same family may have different biological functions. This functional diversity and variability may be related to functional domains and protein localization. The diversity and variability analyzed in this study can help deepen the understanding of the biological functions of the OsPITPs family in plant stress resistance and provide a theoretical basis for studying the role of OsPITPs family proteins in plant disease resistance mechanisms. However, this study still has some limitations. The future research directions can be pursued as follows: (1) Exploring the mechanisms of OsPITPs recognition and binding to specific lipids to understand their important role in signal transduction and lipid transport. (2) Investigating the mechanism of OsPITPs in rice disease resistance to further improve the understanding of rice’s disease resistance mechanisms. (3) Exploring the regulatory mechanisms of the OsPITPs family to understand their role in plant growth, development, and stress response.

## 5. Conclusions

The *OsPITPs* in rice included 1 *OsPITPα*, 27 *OsSEC14* and 2 *OsML* genes, and shown differences in physicochemical properties, gene structure, and protein conservation domain. Interestingly, our predictions through cis-acting elements have shown that *OsPITPs* may be involved in plant disease resistance, containing at least one salicylic acid or methyl jasmonate transient action element. Furthermore, in the early stages of invasion by *M. oryzae*, qRT-PCR result has shown that *OsML-1*, *OsSEC14-3*, *OsSEC14-15*, *OsSEC14-4*, and *OsSEC14-19* are significantly upregulated or downregulated on 12-48 h. This indicates that the rice OsPITP family may participate in the rice resistance to *M. oryzae*. Our data provides a theoretical basis for the further investigation of the rice OsPITP family in plant resistance.

## Figures and Tables

**Figure 1 plants-12-02122-f001:**
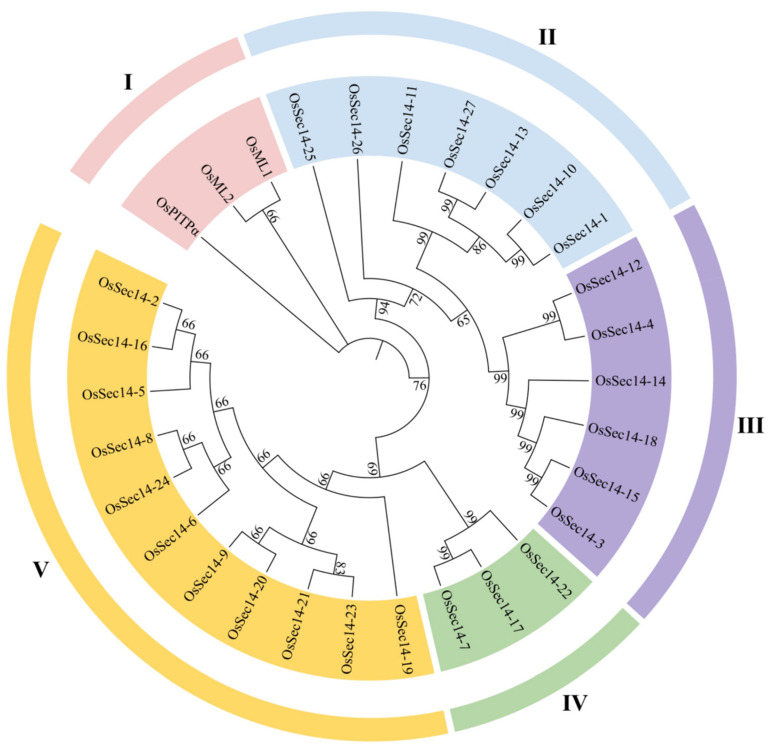
Evolutionary tree analysis of rice PITPs family proteins. The evolutionary tree of the PITPs family in rice was constructed using the maximum likelihood method, and proteins in rice were divided into groups I, II, III, IV, and V based on their phylogenetic relationships.

**Figure 2 plants-12-02122-f002:**
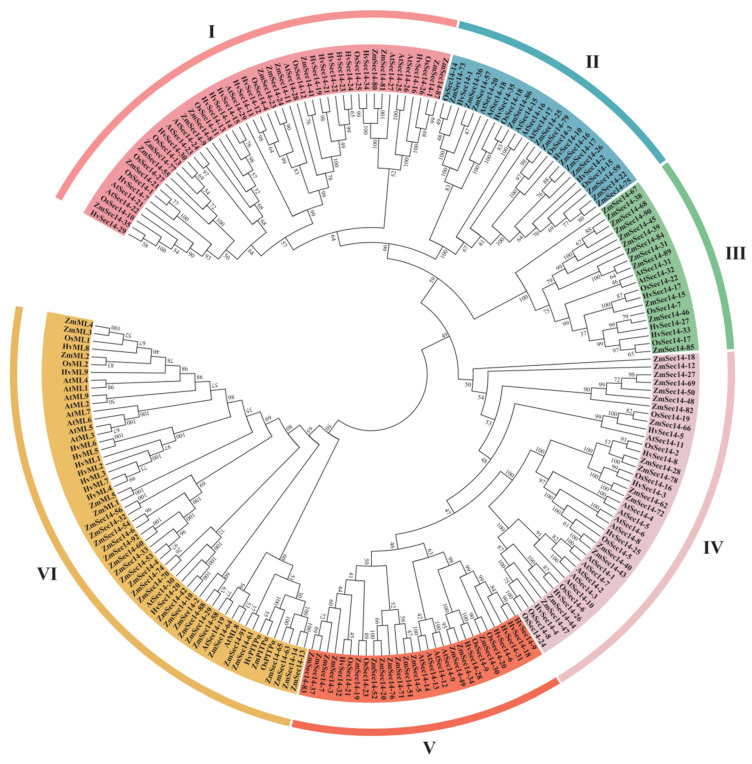
Evolutionary tree analysis of rice PITPs family proteins compared to those of maize, barley, and *Arabidopsis*. Use maximum likelihood method to construct a phylogenetic tree of the PITPs protein family from rice, maize, barley, and *Arabidopsis*. Based on their evolutionary relationships, the family is divided into groups I, II, III, IV, V, and VI.

**Figure 3 plants-12-02122-f003:**
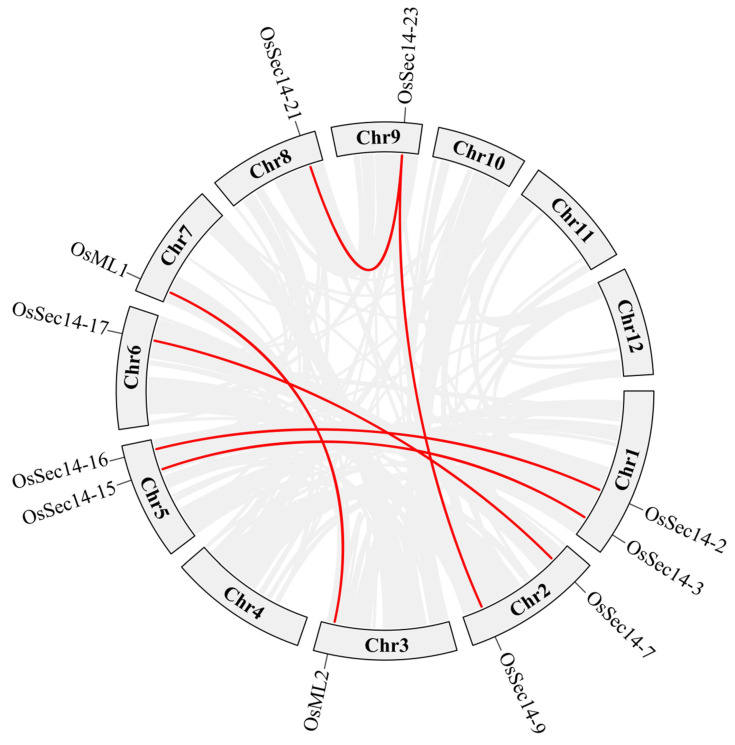
Co-linearity analysis of *OsPITPs*. Use advanced Circos in TBtools to locate all *OsPITPs* sequences to their respective positions in the rice genome.

**Figure 4 plants-12-02122-f004:**
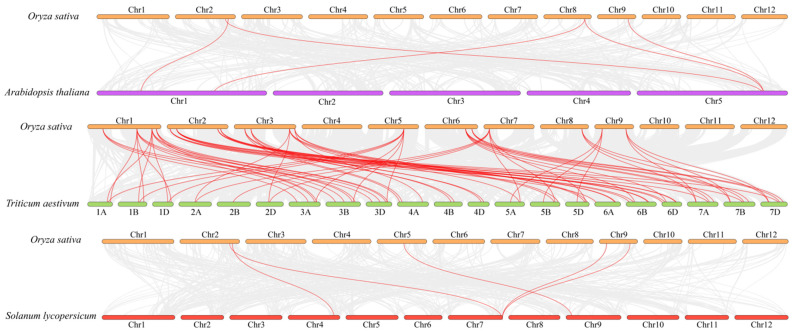
Collinear analysis of the rice PITP rice family across *Arabidopsis thraliana*, *Triticum aestivum*, and *Solanum lycopersicum*.

**Figure 5 plants-12-02122-f005:**
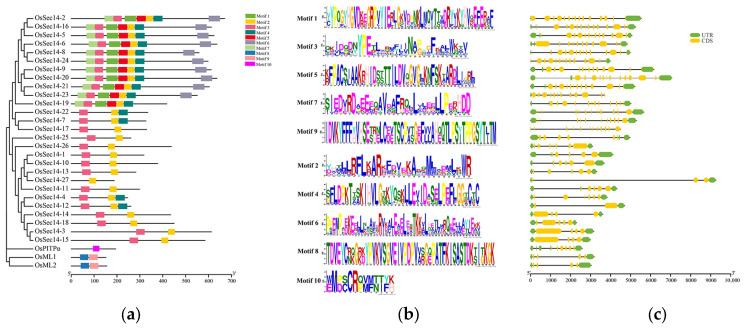
Protein conserved motifs and Gene intron/exon structures of the PITPs Family proteins. (**a**) Conserved motif; (**b**) Conserved motif sequences; (**c**) Exon-intron organizations of *OsPITPs*. Ten conserved motifs labeled with different colors in (**a**) were found in the OsPITPs sequences using the MEME program. In (**c**), the green boxes represent 50 or 30 untranslated regions, yellow boxes represent exons, and black lines represent the introns. The lengths of the exons and introns can be determined by the scale at the bottom.

**Figure 6 plants-12-02122-f006:**
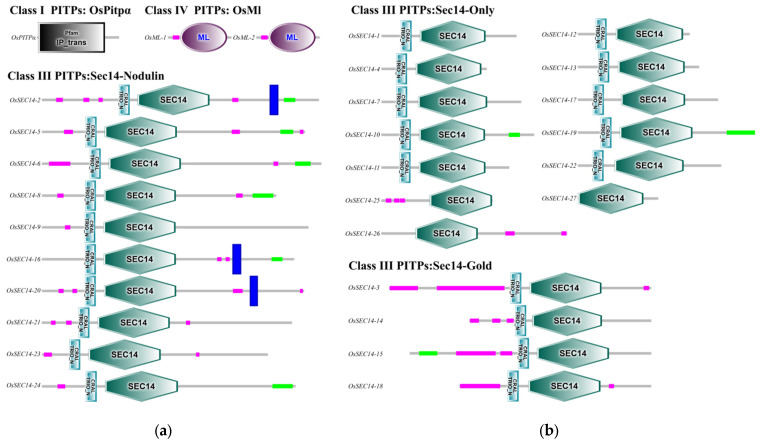
Functional domains of OsPitps. (**a**) OsPitpα, OsMl and Sec14-Nodulin; (**b**) Sec14-Only and Sec14-Gold. The SMART website was used to forecast the functional domain. The black and white gradient color box represents IP_trans domain, purple and white gradient color boxes represent ML domain, blue and white gradient color boxes represent CRAL-TRIO domain, green and white gradient color boxes represent SEC14 domain, purple boxes represent low complexity region, blue boxes represent transmembrane region, and green boxes represent coiled coil region.

**Figure 7 plants-12-02122-f007:**
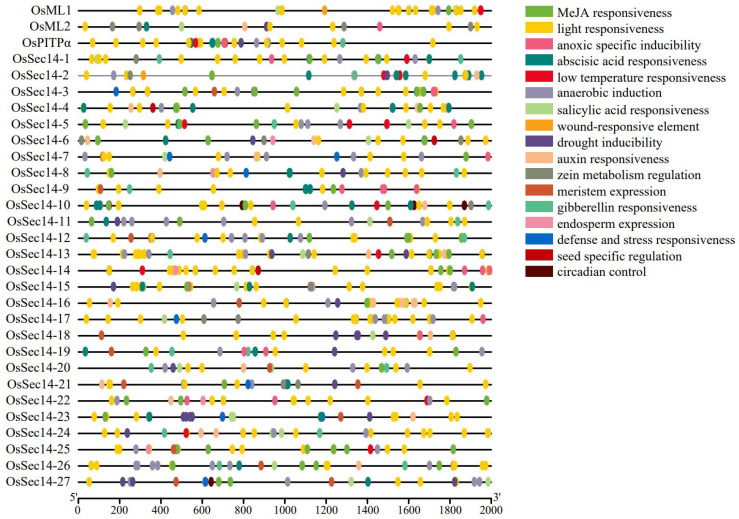
Prediction of cis-elements in the promoter region of the rice PITPs gene family. Elements with similar regulatory functions are displayed in the same color.

**Figure 8 plants-12-02122-f008:**
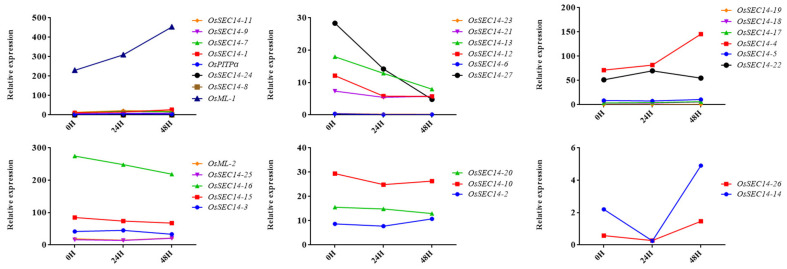
Expression analysis of *OsPITPs* family genes based on transcriptome data. The transcriptome data of Nipponbare (NPB, a susceptible rice variety) after 24 h and 48 h *M. oryzae* infection.

**Figure 9 plants-12-02122-f009:**
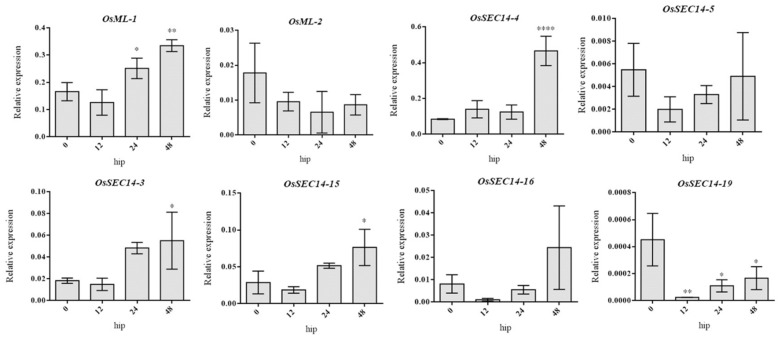
Expression analysis of *OsML-1*, *OsML-2*, *OsSEC-3*, *OsSEC-4*, *OsSEC-5*, *OsSEC-14*, *OsSEC-15*, *OsSEC-16*, and *OsSEC-19* in wild-type plants infected with rice blast fungus at 0 h, 12 h, 24 h, and 48 h. qRT-PCR data were calculated by the methods of 2^−ΔCT^. STDEV is indicated as error bars. All data are the mean of 3 biological replicates. Significant differences are shown by * (*p* < 0.05), ** (*p* < 0.01) and **** (*p* < 0.0001).

**Figure 10 plants-12-02122-f010:**
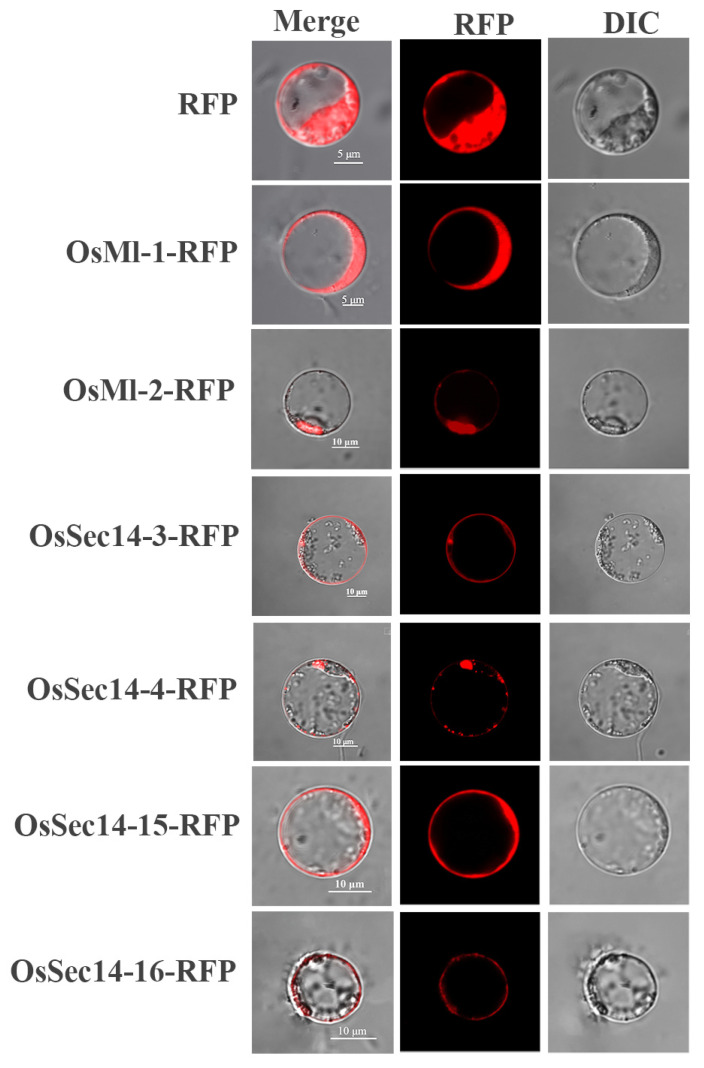
Subcellular localization of OsPITP proteins in rice protoplasts. DIC (Differential interference contrast) represents the natural imaging of rice protoplasts, RFP (Red fluorescence protein) represents the localization of different fusion proteins within cells, and Merge represents the synthetic image of DIC and RFP.

**Table 1 plants-12-02122-t001:** The physicochemical properties of OsPitps.

Protein Name	Gene ID	Amino Acid Number	Molecular Weight	Theoretical pI	Instability Index	Aliphatic Index	GRAVY
OsPitpα	LOC_Os03g17800.1	195	22,111.60	8.40	39.63	87.54	−0.293
OsSec14-1	LOC_Os01g16000.1	318	36,127.60	5.896	41.42	68.71	−0.643
OsSec14-2	LOC_Os01g50616.1	671	75,911.99	9.02	53.38	84.43	−0.556
OsSec14-3	LOC_Os01g65380.1	613	66,457.94	4.65	55.45	74.81	−0.453
OsSec14-4	LOC_Os01g70210.1	247	28,078.39	8.24	42.17	86.11	−0.366
OsSec14-5	LOC_Os02g04020.1	624	71,010.26	8.47	50.20	82.34	−0.527
OsSec14-6	LOC_Os02g04030.1	637	71,116.11	7.19	54.24	70.50	−0.532
OsSec14-7	LOC_Os02g10650.1	327	37,028.52	6.87	39.43	89.94	−0.341
OsSec14-8	LOC_Os02g24430.1	559	63,004.79	7.88	49.62	82.72	−0.404
OsSec14-9	LOC_Os02g48990.1	612	69,750.04	6.49	43.21	79.31	−0.622
OsSec14-10	LOC_Os02g51610.1	378	42,355.55	5.16	50.09	70.77	−0.636
OsSec14-11	LOC_Os03g51430.1	299	34,667.65	8.55	51.96	78.90	−0.528
OsSec14-12	LOC_Os03g63370.1	261	30,159.22	8.20	46.17	77.39	−0.621
OsSec14-13	LOC_Os05g18470.1	283	32,973.69	8.88	48.97	71.38	−0.616
OsSec14-14	LOC_Os05g27820.1	435	47,828.78	8.27	44.09	83.29	−0.183
OsSec14-15	LOC_Os05g35460.1	585	63,948.04	4.59	48.61	77.85	−0.457
OsSec14-16	LOC_Os05g46720.1	613	70,003.75	8.34	51.93	73.03	−0.621
OsSec14-17	LOC_Os06g40510.1	330	37,299.38	5.67	37.85	92.70	−0.377
OsSec14-18	LOC_Os06g45990.2	450	49,793.09	4.83	52.01	76.80	−0.517
OsSec14-19	LOC_Os07g27310.1	418	47,494.42	8.55	45.03	88.90	−0.427
OsSec14-20	LOC_Os08g25310.1	637	73,245.23	8.91	62.16	72.70	−0.669
OsSec14-21	LOC_Os08g38850.1	604	69,256.46	6.86	55.89	77.95	−0.595
OsSec14-22	LOC_Os09g08390.1	335	37,958.67	7.65	36.43	85.55	−0.331
OsSec14-23	LOC_Os09g30330.1	551	63,232.14	7.29	63.43	75.34	0.507
OsSec14-24	LOC_Os10g03400.2	598	67,600.15	8.49	54.00	77.01	−0.484
OsSec14-25	LOC_Os02g21630.1	261	29,758.36	8.86	44.11	86.55	−0.248
OsSec14-26	LOC_Os03g11950.1	438	48,675.95	9.95	50.69	78.45	−0.219
OsSec14-27	LOC_Os05g18294.1	188	21,704.77	6.39	53.41	89.20	−0.384
OsMl-1	LOC_Os07g06590.1	152	16,196.64	5.32	36.31	89.87	0.215
OsMl-2	LOC_Os03g57420.1	156	16,946.55	5.66	40.16	96.86	0.238

**Table 2 plants-12-02122-t002:** The Subcellular localization prediction of OsPitps.

Protein Name	CELLO	Plant-mPLoc	Protein Name	CELLO	Plant-mPLoc
OsPitpα	Nucleus	Nucleus	OsSec14-15	Nucleus	Cytoplasm
OsSec14-1	Nucleus	Cytoplasm	OsSec14-16	Chloroplast	Cytoplasm
OsSec14-2	Nucleus	Cytoplasm	OsSec14-17	Nucleus	Cytoplasm
OsSec14-3	Nucleus	Nucleus	OsSec14-18	Nucleus	Cytoplasm
OsSec14-4	Cytoplasm	Cytoplasm	OsSec14-19	Mitochondrio	Cytoplasm
OsSec14-5	Chloroplast	Cytoplasm	OsSec14-20	Nucleus	Cytoplasm
OsSec14-6	Chloroplast	Cytoplasm	OsSec14-21	Chloroplast	Cytoplasm
OsSec14-7	Cytoplasm	Cytoplasm	OsSec14-22	Cytoplasm	Cytoplasm
OsSec14-8	Cytoplasm	Cytoplasm	OsSec14-23	Mitochondrio	Cytoplasm
OsSec14-9	Chloroplast	Cytoplasm	OsSec14-24	Chloroplast	Cytoplasm
OsSec14-10	Cytoskeleton	Cytoplasm	OsSec14-25	Cytoplasm	Cytoplasm
OsSec14-11	Nucleus	Cytoplasm	OsSec14-26	Cytoplasm	Cytoplasm
OsSec14-12	Cytoplasm	Cytoplasm	OsSec14-27	Cytoplasm	Cytoplasm
OsSec14-13	Cytoplasm	Cytoplasm	OsMl-1	Cytoplasm	Nucleus
OsSec14-14	Chloroplast	Cytoplasm	OsMl-2	Extracellular	Nucleus

**Table 3 plants-12-02122-t003:** The list of qRT-PCR primers used in this study.

Primer Name	Primer Sequence (5′-3′)	Exon-Exon Junction	qRT-PCR Products (bp)
QRT-OsACTIN-F	GGACTCTGGTGATGGTGTCAGCCA	Yes	243
QRT-OsACTIN-R	GAGCTGGTCTTGGCAGTCTCCA		
QRT-OsML-1-F	GTGGTGACTTCTTGGTAG	Yes	76
QRT-OsML-1-R	TTCATGGTGATGGTGTAAG		
QRT-OsML-2-F	CACGAACAGACTTTACCA	Yes	75
QRT-OsML-2-R	GTTGCCGTCATCTAGTAG		
QRT-OsSEC14-3-F	ACCGTTGAGATTCCTGTC	Yes	89
QRT-OsSEC14-3-R	GTGAACTCTGCTCCGTAG		
QRT-OsSEC14-4-F	TGGAAGATGATCTACCCTTTC	Yes	103
QRT-OsSEC14-4-R	TATCGTCGATCTCCTGGT		
QRT-OsSEC14-5-F	TTCAAGATAAGCCTTCAG	Yes	125
QRT-OsSEC14-5-R	GATAAGAGCCTCATGTAG		
QRT-OsSEC14-15-F	AACTGTTGAGATTCCTGCTA	Yes	90
QRT-OsSEC14-15-R	GTGAACTCTGCACCGTAG		
QRT-OsSEC14-16-F	GAGATGCCATTTGAAAAG	Yes	183
QRT-OsSEC14-16-R	CTAGAAGCAGAATTTCTTT		
QRT-OsSEC14-19-F	TACTACAGAAGCAGAATG	Yes	107
QRT-OsSEC14-19-R	TCACTTCAAGATTAGGAC		

## Data Availability

All data generated or analyzed during this study are included in this article. The sequences of 30 OsPITPs downloaded from the rice genome are accessible via the following link http://rice.uga.edu/cgi-bin/gbrowse/rice/, accessed on 7 February 2023.

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
