# Peer review of "Identification and Expression Analysis of Phosphatidylinositol Transfer Proteins Genes in Rice"

_plants, 2023, doi:10.3390/plants12112122_

Round 1

Reviewer 1 Report

Authors performed interesting study concerning Expression Analysis of Phosphatidylinositol 2 Transfer Proteins Genes in Rice.

Some issues should be correceted before the publication:

1.      Figure 8 – not all genes present in figure description are on the figure, try to include all genes.

2.      Figure 8 and 9; there are significant differences in gene expression based on transcriptomic data and RT-PCR. Typically expression based on transcriptomic results is within tens or hundreds while the RT-PCR show values below 1 for the same genes. Authors should explain such differences.

3.      Figure 10- describe what means DIC.

4.       Describe transcriptomic test in material and methods, include information what are controls. Authors  may try to use suggestion from this article: A Guide for Designing and Analyzing RNA-Seq Data | Springer Nature Experiments

5.      Describe more precisely RT-PCR experiment in material and metod section; method of RNA extraction, metod of RNA quality assessment, describe how the traces of genomic DNA were removed, amount of RNA per analysed sample, name of reference gene, length of PCR products for tested and reference gene, sequences of primers, details of RT and PCR reaction, name of PCR equipment, software used to acquire raw RT-PCR data, citation of 2−ΔΔCT method.  Authors may consult such article:

The MIQE guidelines: minimum information for publication of quantitative real-time PCR experiments - PubMed (nih.gov)

6.      Rewrite the discussion section. Discussion should concentrate on Authors own results and discuss them in context of their similarities and dissimilarities with related/other Authors studies. Now it is rather related to Introduction.

 Minor comments:

Correct typographical errors in following lines (lack of space, additional dots): 65,79,84,87,93,100,104,125,290,292,357. Check the entire text.

Line 275- write Protein not protein.

318- write kilogram not Kilogram

535,541- Latin names of organisms write in italics, check the en tire text.

In my opinion only minor editing of English language is required. Authors should correct typographical errors described above.

Author Response

1.1.Figure 8 not all genes present in figure description are on the figure, try to include all genes..

Answer: Sorry, when we were working on the chart, we mistakenly named two items with the same name. It has been modified in Figure 8.

  1. Figure 8 and 9; there are significant differences in gene expression based on transcriptomic data and RT-PCR. Typically expression based on transcriptomic results is within tens or hundreds while the RT-PCR show values below 1 for the same genes. Authors should explain such differences..

Answer: When analyzing transcriptome data, the company used a different calculation method than QPCR. The RNA-seq calculation method measures the expression level of the target gene (sequencing read count) relative to the total RNA expression level of the sample (total sequencing read count), while the QPCR calculation method measures the expression level of the target gene (cycle count) relative to the expression level of the housekeeping gene (cycle count). This caused some data differences.

At the same time, transcriptome analysis is a high-throughput technology that looks at the overall transcription levels of all genes and can detect low-expression genes, but there may be errors and false-positive signals. QRT-PCR, on the other hand, is a more precise quantification method that can selectively detect one or several genes and can detect very low expression levels. Therefore, the same gene may show different trends.

  1. Figure 10-describe what means DIC.

Answer: apologize for not explaining clearly. DIC is an imaging mode in confocal microscopy. The full name is Differential Interference Contrast (DIC) technology, also known as Nomarski microscopy. This technology converts the phase image of the sample into an intensity image by using a phase shifter, polarizer, and optical devices, thus obtaining high-resolution and high-contrast 3D images. It has been marked in Figure 10

4.Describe transcriptomic test in material and methods, include information what are controls. Authors may try to use suggestion from this article: A Guide for Designing and Analyzing RNA-Seq Data | Springer Nature Experiments

Answer: Thank you for your suggestion. We have added it to the Materials and Methods section.

We inoculated NBP plants with the G11 strain and collected samples at 0, 24, and 48 hours post-infection. Control samples were also collected from uninfected NBP plants at the same time points. Subsequently, all samples were flash-frozen in liquid nitrogen and sent to BioMarker Technologies for transcriptome sequencing analysis.

5.Describe more precisely RT-PCR experiment in material and metod section; method of RNA extraction, metod of RNA quality assessment, describe how the traces of genomic DNA were removed, amount of RNA per analysed sample, name of reference gene, length of PCR products for tested and reference gene, sequences of primers, details of RT and PCR reaction, name of PCR equipment, software used to acquire raw RT-PCR data, citation of 2−ΔΔCT method.Authors may consult such article:The MIQE guidelines: minimum information for publication of quantitative real-time PCR experiments - PubMed (nih.gov)

Answer: We have made modifications to the Materials and Methods section (3.9 RNA Extraction and qRT-PCR) as requested, and the primer sequences can be found in table3.We also apologize for having discovered an error. We used '2−ΔCT' to calculate the relative expression levels, and we have made a new reference to this in the Methods section.

6.Rewrite the discussion section. Discussion should concentrate on Authors own results and discuss them in context of their similarities and dissimilarities with related/other Authors studies. Now it is rather related to Introduction.

Answer: We have rewritten the Discussion section based on the following five points: 1. Elaborating on the research conclusions; 2. Analyzing and interpreting the results; 3. Comparing and contrasting with other studies; 4. Discussing the significance and application of the findings; and 5. Proposing future research directions.

Minor comments:

1.Correct typographical errors in following lines (lack of space, additional dots): 65,79,84,87,93,100,104,125,290,292,357. Check the entire text.

Answer: We have revised and checked the entire text.

  1. Line 275- write Protein not protein.

Answer: This has been modified

318- write kilogram not Kilogram

Answer: This has been modified

535,541- Latin names of organisms write in italics, check the en tire text.

Answer: This has been modified and checked the entire text 

Reviewer 2 Report

The manuscript "Identification and expression analysis of phosphatidylinositol 2 transfer proteins genes in rice" identified and preliminarily analyzed 30 PITPs genes in rice. The experiments are well-designed and performed, and the conclusions are appropriate to the results presented. I have some minor suggestions.  

Line 169, from this line through the paragraph, double-check the number of each protein property mentioned is correct. E.g., the proteins with pI>7.5 from Table 1 should be 16 if I count correctly.  

Line 194 you are talking about OsPitpa homologous in maize, barley, and Arabidopsis. So, you can change “OsPitpα proteins” to “OsPitpα protein homologs”. 

Line 207, the current phylogeny result in Figure 2 doesn’t convince me because some bootstrap values are too low. For example, the value on the node that derives group I, II, III, IV, and V is 3, which means this big clade only shows up 3 times when performing 100 times of resampled ensemble of trees.

Line 363, what is DIC in the legend? 

Line 513, duplicate paragraphs from this line through 603. 

Author Response

  1. Line 169, from this line through the paragraph, double-check the number of each protein property mentioned is correct. E.g., the proteins with pI>7.5 from Table 1 should be 16 if I count correctly.  

Answer: We are sorry to make mistakes, they have been modified

2.Line 194 you are talking about OsPitpa homologous in maize, barley, and Arabidopsis. So, you can change “OsPitpα proteins” to “OsPitpα protein homologs”.

Answer: Thankyou,, We have modified it

  1. Line 207, the current phylogeny result in Figure 2 doesn’t convince me because some bootstrap values are too low. For example, the value on the node that derives group I, II, III, IV, and V is 3, which means this big clade only shows up 3 times when performing 100 times of resampled ensemble of trees.

Answer: I'm sorry, this is indeed an issue. We have conducted a new phylogenetic tree analysis.

  1. Line 363, what is DIC in the legend?

Answer: apologize for not explaining clearly. DIC is an imaging mode in confocal microscopy. The full name is Differential Interference Contrast (DIC) technology, also known as Nomarski microscopy. This technology converts the phase image of the sample into an intensity image by using a phase shifter, polarizer, and optical devices, thus obtaining high-resolution and high-contrast 3D images. It has been marked in Figure 1

  1. Line 513, duplicate paragraphs from this line through 603.

Answer: I apologize, the required changes have been implemented.

Round 2

Reviewer 1 Report

Authors corrected all errors and answered all questions. I have no other comments.

Minor editing of English language required.